# High-Risk Human Papillomavirus and Epstein–Barr Virus Coinfection: A Potential Role in Head and Neck Carcinogenesis

**DOI:** 10.3390/biology10121232

**Published:** 2021-11-26

**Authors:** Rancés Blanco, Diego Carrillo-Beltrán, Alejandro H. Corvalán, Francisco Aguayo

**Affiliations:** 1Programa de Virología, Instituto de Ciencias Biomédicas (ICBM), Facultad de Medicina, Universidad de Chile, Santiago 8380000, Chile; rancesblanco1976@gmail.com (R.B.); diegocb17@hotmail.com (D.C.-B.); 2Advanced Center for Chronic Diseases (ACCDiS), Pontificia Universidad Católica de Chile, Santiago 8320000, Chile; acorvalan@uc.cl; 3Universidad de Tarapacá, Arica 1000000, Chile

**Keywords:** Epstein–Barr virus, human papillomavirus, head and neck cancer

## Abstract

**Simple Summary:**

A subset of carcinomas that arise in the head and neck region show a viral etiology. In fact, a subgroup of oropharyngeal cancers are caused by some types of human papillomavirus (HPV), so-called high-risk (HR)-HPVs, whereas undifferentiated nasopharyngeal carcinomas are etiologically related to Epstein–Barr virus (EBV). However, studies have reported the presence of both HR-HPV and EBV in some types of head and neck cancers. In this review, we discuss the potential contribution and role of HR-HPV/EBV coinfection in head and neck carcinogenesis, as well as the mechanisms that are potentially involved. In addition, HR-HPV/EBV interaction models are proposed.

**Abstract:**

High-risk human papillomaviruses (HR-HPVs) and Epstein–Barr virus (EBV) are recognized oncogenic viruses involved in the development of a subset of head and neck cancers (HNCs). HR-HPVs are etiologically associated with a subset of oropharyngeal carcinomas (OPCs), whereas EBV is a recognized etiological agent of undifferentiated nasopharyngeal carcinomas (NPCs). In this review, we address epidemiological and mechanistic evidence regarding a potential cooperation between HR-HPV and EBV for HNC development. Considering that: (1) both HR-HPV and EBV infections require cofactors for carcinogenesis; and (2) both oropharyngeal and oral epithelium can be directly exposed to carcinogens, such as alcohol or tobacco smoke, we hypothesize possible interaction mechanisms. The epidemiological and experimental evidence suggests that HR-HPV/EBV cooperation for developing a subset of HNCs is plausible and warrants further investigation.

## 1. Introduction

Head and neck cancers (HNCs) encompass tumors of a variety of subsites, including the lips, oral cavity, nasopharynx, oropharynx, hypopharynx, larynx, and salivary glands. Head and neck squamous cell carcinomas (HNSCCs) constitute more than 90% of all HNCs [1]. These malignancies were the sixth leading cancer worldwide, with 931,931 new cases in 2020, excluding non-melanoma skin neoplasms. Additionally, HNCs represented the seventh highest cause of cancer-related deaths in the same year, with 467,125 deaths [2]. Alcohol consumption and tobacco smoking are the two most important risk factors for HNC development. Infection with oncogenic viruses, such as the human papillomavirus (HPV) or the Epstein–Barr virus (EBV), have been reported to be associated with the development and progression of these neoplasms. In fact, high-risk (HR)-HPV infection is a risk factor for developing a subset of oropharyngeal tumors, including the tonsils and the base of the tongue [3]. Similarly, EBV infection is related to the development of nasopharyngeal (NPC) and salivary gland cancers [4]. However, neither HR-HPV nor EBV infection alone is a sufficient condition for tumorigenesis. In most immunocompetent subjects, HPV infection is ultimately controlled [5], whereas EBV is present during the lifetime of the individual without any clinical manifestation, all of which suggesting that additional cofactors are required for promoting HR-HPV or EBV-driven cancers. Indeed, some studies have reported the simultaneous presence of HR-HPV and EBV in HNCs [6,7,8], proposing that HR-HPV/EBV coinfection could play a fundamental role in the development of these malignancies [6,9,10]. However, potential mechanisms involved in such HPV/EBV cooperation need further clarification. Here, we review the current literature regarding HR-HPV/EBV coinfection in HNSCCs, as well as its potential contribution to carcinogenesis and the progression of these tumors. Finally, we propose a mechanism by which HR-HPV-mediated alterations facilitate EBV infection in epithelial tissues.

## 2. Human Papillomavirus and Epstein–Barr Virus Tropism

Epithelial tissues are susceptible and permissive to both HPV (cutaneous and mucosal) [11] and EBV (mucosal) infections [12]. Basal epithelial cells are susceptible to HPV infection, although only upper highly differentiated cells in the stratified epithelium are permissive [13,14]. HPV entry into epithelial cells requires the involvement of heparan sulphate proteoglycans (HSPGs) for initial tethering to basal cells [15]. Integrins (α6 and α3) and CD151 are also required to form an “entry receptor complex” with annexin A2, CD63, and EGFR/GFR (reviewed in [16]). Interestingly, multiple studies have reported that HPV can regulate integrin levels. For instance, Werner et al. (2011) reported increased levels of αv, α5, β1, β4, and β6 integrins in HPV-positive cervical cells when compared with non-tumor cells, which was attributable to HPV infection [17]. Overexpression of HPV16 E6*, the smaller splice isoform of the E6 oncogene, increased the levels of the β1 integrin in cervical cancer cells [18]. A 2.44-fold increase in the expression of the ITGB5 gene, which encodes the β5 integrin subunit, was reported in cervical cells upon L2 overexpression [19]. In contrast, it was reported that HPV38 E6/E7 proteins downregulate the expression of αvβ8 integrin in human keratinocytes [20].

EBV exhibits a similar epithelial tropism, although EBV cell entry occurs through both apical and basolateral routes in mucosal epithelia [21], and this virus shows an additional and preferential tropism for B cells, which constitute the EBV reservoir [22]. EBV lytic infections occur in highly differentiated cells, as demonstrated by Temple et al. (2014) [23]. EBV entry to epithelial cells is mediated by the ephrin receptor tyrosine kinase A2 (EphA2). Furthermore, the increased expression of EphA2 leads to a more efficient EBV infection of epithelial cells [24,25]. Of note, elevated EphA2 expression was associated with progression and poor prognosis in HNSCC patients [26]. Notably, some EBV-encoded proteins (e.g., LMP1) can regulate the expression of the Eph family [27]. Studies by Chesnokova et al. suggested that the interaction of EBV glycoproteins gH/gL with αvβ5, αvβ6, and αvβ8 integrins is associated with EBV fusion with gastric carcinoma cells [28,29], though Chen et al. (2018) reported a lack of effect of these surface integrins on EBV entry in HEK293 cells [24]. Interestingly, the expression of αvβ5 integrin was significantly increased in oral and laryngeal SCCs when compared to normal counterparts [30,31]. In addition, αvβ6 integrin was evidenced in almost all HNCs, including oral, oropharyngeal, hypopharyngeal, laryngeal, and nasopharyngeal carcinomas, but not in normal oral and tongue mucosa [32]. The expression of αvβ8 integrin was also found to be increased in laryngeal carcinoma compared with normal tissues [31]. A comparison between HPV and EBV tropism is shown in Table 1.

## 3. Epidemiology of HPV/EBV Coinfection in HNSCCs

As in epidemiological studies, the cellular location of viruses is generally not considered, and the term “copresence” is assumed when HPV and EBV are both detected in tumors. However, copresence in tumors is not synonymous with coinfection, which occurs when both HPV and EBV infects the same epithelial cell. Molecular methodologies, such as a polymerase chain reaction (PCR), cannot determine the coinfection or coexpression of viral products, although this test is frequently used in epidemiological studies. HPV/EBV copresence has been evidenced in 31.2% of HNCs, which includes larynx, lip, and nasopharynx malignancies [9]. Similarly, Drop et al. found HPV/EBV copresence in 34.1% of oral cavity, oropharyngeal, and laryngeal tumors [6], and EBV infection was significantly associated with HPV16, 18, 45, and 58 presence in HNSCCs [10]. Remarkably, the expression of both LMP1 and E6 oncoproteins in HNCs was associated with an increased cell invasiveness and advanced tumor stage [9,10]. HPV/EBV copresence was also associated with alcohol abuse and an increased histological grade of HNCs, although it was more frequent in early-stage disease [6]. In the following subsections, we will analyze the HPV/EBV copresence in HNCs from different anatomical sites.

### 3.1. Oral Cavity

More than 90% of oral tumors are squamous cell carcinomas (OSCCs), which arise from the mucosal epithelium lining the lips, anterior tongue, gingivae, floor of mouth, palate, and other areas of the mouth [38]. In 2020, there were an estimated 377,713 new cases and 177,757 deaths from lip and oral cavity tumors worldwide [2]. Tobacco smoking and alcohol consumption are the two most important risk factors associated with oral cancer development. Additionally, the involvement of some viral infections has also been suggested in oral carcinogenesis (reviewed in [39]). HPV infection in OSCCs ranges between 6% and 58% worldwide [40,41,42]. A meta-analysis including 62 studies and 4852 OSCCs found that 38.1% of cancers contained HPV DNA [43]. It was also discovered that HPV infection increased the probability of OSCC development in the Chinese population (OR 12.7, 95% CI: 8.0–20.0) by more than 12 times [44]. However, a recent meta-analysis published by Melo et al. (2021) found E6/E7 mRNA expression in only 4.4% of OSCCs [45]. HPV16 and 18 are the most frequent HPV genotypes detected in these tumors [45,46]. In addition, the presence of HPV16 has been related to an increased histological grade of oral tumors [47]. Meanwhile, EBV infection has been detected in 25.9% to 82.5% of OSCCs [48,49,50]. EBNA2 and LMP1 were also detected in 50.2% and 10.7% of OSCCs, respectively [51]. In particular, the expression of LMP1 was increased in OSCCs when compared to normal mucosa and oral leukoplakia [52]. She et al. (2017) conducted a meta-analysis that included 13 studies published between 1995 and 2016. In this report, OSCC development was shown to be 5 times more likely to occur in the presence of EBV (OR 5.03, 95% CI: 1.80–14.01) [53]. Another meta-analysis also found an association between EBV infection and OSCC. In fact, this study reported that the probability of OSCC development was 2.5 (OR 2.5, 95% CI: 1.2–5.36) times increased in EBV-positive cases [54]. Regarding HPV/EBV coinfection, this was evidenced in 6.5–37.5% of OSCCs [6,10,55,56]. In one study, which included 155 OSCCs from eight countries, HPV/EBV was detected in 21% of cases, making it the most frequent coinfection in these tumors [55]. The coexpression of HR-HPV E6 and LMP1 was also demonstrated in OSCCs by immunohistochemistry (IHC) [10]. In particular, HPV16 and EBV copresence was detected in 5.6% of cases [57]. Moreover, EBV infection was associated with HR-HPV16, 18, 31, 35, 45, 51, and 58 [10]. Conversely, an absence of EBV/HPV copresence has also been reported in oral cancers [58,59].

### 3.2. Nasopharynx

Nasopharyngeal carcinoma (NPC) is a HNC with an estimated 133,354 new cases and 80,008 deaths in 2020 [2]. NPC usually exhibits a heterogeneous racial and geographic distribution. Its frequency is high in Southern China, intermediate in Southeast Asia and Northern Africa, and low in most areas of the word, with 15–20.5 and <1 per 100,000 habitants, respectively [60]. According to the World Health Organization (WHO) pathologic classification, NPCs are divided into keratinizing squamous cell carcinoma (WHO type I) and non-keratinizing carcinoma (WHO types II and III) [61]. In endemic areas, the keratinizing subtype is relatively rare, whereas the non-keratinizing subtype can reach more than 99% of total cases [62,63]. Diverse factors, such as salted fish consumption, smoking, alcohol consumption, air pollution exposure (e.g., NO2), and genetic susceptibility, are among the risk factors related to NPC development [64,65,66]. EBV is a well-established etiological agent of undifferentiated (non-keratinizing) NPC [67,68,69], which is detected in almost 100% of cases [70,71]. Overall, 57.6% to 100% of NPCs are EBV-positive [72,73,74,75,76]. A meta-analysis reported that EBV prevalence was higher in WHO type II/III NPC compared to WHO type I tumors (83.2% vs. 21.3%) [77]. Importantly, increased levels of plasma EBV DNA are related to a more aggressive biological behavior of this cancer [78,79,80,81]. However, an association was reported between the EBV viral load and a better prognosis in NPC patients [82]. The expression of LMP1, a latent EBV protein, was associated with the TNM stage and lymph node metastasis in NPC patients [61]. In a meta-analysis conducted by Zhao et al. (2012), the occurrence of metastasis in NPC was 1.98 times increased in the presence of LMP1 (OR 1.98, 95% CI: 1.38–2.83) [83]. On the other hand, Bam HI rightward Frame 1 (BARF1), which is considered an exclusively epithelial oncoprotein, is detected in 69–100% of NPCs [84]. Regarding HPV infection, the pooled prevalence worldwide was 21%, and it increased 4.77-fold in tumor samples compared to control cases [85]. HPV16 and HPV18 are the most common genotypes detected in HPV-positive NPCs [77]. Of note, an increased prevalence of HR-HPV was evidenced in WHO type I NPC when compared with WHO type II/III cases (39.9% vs. 23.3%) [77]. Moreover, patients with HPV-positive tumors displayed a decreased overall survival, progression-free survival, and locoregional control when compared to exclusively EBV positive patients [86]. In the meta-analysis conducted by Tham et al. (2020), an increased frequency of HPV/EBV copresence was found in WHO type I NPCs when compared with WHO type II/III tumors (7.6% vs. 1.0%) [77]. In addition, HPV/EBV copresence was evidenced mainly in NPC-endemic regions, ranging from 15% to 47.7% of the tumors [8,87,88]. Of note, about 80% of HPV16 or HPV18-positive NPCs were also positive for EBV DNA [87]. The overexpression of p16, a surrogate biomarker of oncogenic HPV infection, was associated with an improved progression-free survival and locoregional control in EBV-positive NPCs [89]. However, in non-endemic areas, HPV/EBV coinfection is less common. Indeed, some authors reported that HPV and EBV infections are mutually exclusive in populations with a low NPC incidence [86,90].

### 3.3. Oropharynx

An estimated 98,412 new cases and 48,143 cancer deaths worldwide were attributable to oropharyngeal tumors in 2020 [2]. Most oropharyngeal tumors are squamous cell carcinomas (OPSCC), which arise from epithelial cells lining the oropharynx [91]. Etiologically, two different types of OPSCCs have been described: HPV-driven and HPV-negative carcinomas, characterized by a distinct biological and clinical behavior [92]. In fact, patients with HPV-driven OPSCCs usually show a better prognosis compared to those with HPV-negative tumors [93,94]. Furthermore, patients with HPV16-driven OPSCC showed an increased overall survival rate when compared to patients with non-HPV16-positive carcinomas [95]. HPV-negative OPSCCs are mainly related to other risk factors, such as tobacco smoke and alcohol consumption [96]. The prevalence of HPV in OPSCC displays a geographic heterogeneity. For instance, a systematic review showed HPV positivity ranging from 18% to 65% in OPSCC patients from European countries [97]. In addition, Mariz et al. (2020) found a 44.8% pooled prevalence of HPV-driven OPSCC, with higher percentages found in New Zealand (74.5%), Sweden (70.0%), and Denmark (61.7%) when compared with the Netherlands (30.3%), Germany (25.0%), and Brazil (11.1%) [98]. Regarding the oropharynx subsites, HPV infection is more frequent in OPSCCs from the base of the tongue (40%) and tonsils (56%) when compared with other sites, such as the soft palate (12%) and posterior wall (19%) [99]. HPV16 is the most frequent genotype detected in OPSCC, reaching 88% of HPV-positive cases [100]. However, other authors reported significant differences in the prevalence of HPV16-positive OPSCCs in the United States (59.3%) and Europe (31.1%) in comparison with Brazil (4.1%) [101]. Other genotypes, such as HPV18, HPV33, HPV35, and HPV58, have also been detected, although only in a small percentage of cases [102]. Interestingly, EBV infection was evidenced in 53.3–86% of OPSCCs [56,103], suggesting its contribution to carcinogenesis. In normal oropharyngeal tissues, an absence of HPV/EBV coinfection was evidenced [7]. Conversely, the copresence of HPV and EBV was detected in 20% and 25% of tonsillar and base of tongue SCCs, respectively, while the absence of HPV/EBV coinfection in soft palate tumors was reported [7]. Similarly, Drop et al. identified HPV/EBV copresence in 28.6% of oropharyngeal tumors [6], whereas other authors only found HPV/EBV infections in 8–9% [56]. Additionally, Broccolo et al. reported HPV infection in 25% of OPSCCs, whereas EBV infection was found in 45% of tumors. However, coinfection was only evidenced in 4% of OPSCCs [58].

### 3.4. HPV and EBV Infection in Other HNSCCs

HNCs also include tumors arising from the hypopharynx, larynx, paranasal sinuses, nasal cavity, and salivary glands. However, data regarding HPV/EBV coinfection in these neoplasms are scarce. HPV/EBV coinfection was reported in 25% and 33.3% of hypopharyngeal and paranasal sinus carcinomas, respectively [104]. In addition, the copresence of HPV and EBV was evidenced in 12.8% [105] and 21.4% [6] of laryngeal carcinomas. Of interest, a three-fold increase was reported in the invasiveness properties of HPV+/EBV+ NOK and FaDu hypopharyngeal cells in the presence of lysophosphatidic acid (LPA) compared to HPV-/EBV- and HPV-/EBV+ cells [7]. Table 2 and Table 3 show meta-analysis data for HPV and EBV detection in HNSCCs.

## 4. Possible Mechanisms of HR-HPV/EBV Cooperation

### 4.1. HPV Facilitates EBV Entry in Epithelial Cells

The cluster of differentiation 21 (CD21), also known as the complement receptor type 2 (CR2), is considered to be the major attachment receptor for EBV in B cells [114]. However, CD21 is variably detected in epithelial cells. Interestingly, an increased CD21 expression was detected in the tonsil epithelium, comparable to what was detected in peripheral B cells, Burkitt lymphoma cells [115], and OSCCs [116]. Furthermore, a CD21 mRNA increase is associated with the grade of oral epithelial dysplasia, and correlates with the presence of EBER1 transcripts [116]. Additionally, epithelial cells ectopically expressing CD21 become susceptible and permissive to EBV infection [117,118]. In fact, both human keratinocytes and skin cancer cells were susceptible to EBV infection after CD21 gene transfection, whereas un-transfected cells remained EBV-negative [117]. CD21 gene-transfected epithelial cells also increase the expression of EBERs and EBNA1, along with a limited LMP1 expression [117,118], which resembles the EBV latency program detected in NPCs [119,120]. Low levels of CD21 expression are sufficient for leading to EBV infection in human epithelial cells [121]. Taken together, these findings suggest a role of CD21 in EBV entry into epithelial cells. Conversely, an absence of CD21 expression was evidenced in oral tongue and mucosa, soft palate, and the uvula [115], and a decreased CD21 expression was found in NPCs when compared to normal nasopharynx [122]. Interestingly, CD21 levels were associated with HPV and EBV infection in tonsillar and base of tongue carcinomas [7]. A high level of CD21 was found in HPV+/EBV+ NPCs, when compared to EBV-/HPV- tumors (*p* = 0.0194), although no difference was observed between HPV+/EBV+ NPCs and those harboring EBV or HPV single infections [7]. Furthermore, increased levels of EphA2, the epithelial EBV receptor, was evidenced in CIN and cervical carcinomas when compared to normal cervical epithelium, and also correlated with CDK6 protein expression [123]. In addition, a 15.28-fold increase in EphA2 receptor expression was observed in non-tumor cervical cells transfected with the HPV18 E2 gene [124]. Similarly, transfection with the HPV18 E6 gene resulted in a 3.2-fold increase in EphA2 gene expression in esophageal cancer cells [125]. These facts suggest that HPV could potentially facilitate EBV entry into head and neck epithelial cells by promoting increased levels of proteins involved in this process. Interestingly, the fact that persistent viruses promote the expression of molecules involved in the entry of another virus is not new. In fact, it was recently reported that EBV infection promotes increased levels of the angiotensin-converting enzyme 2 (ACE2) receptor, facilitating SARS-CoV-2 entry to epithelial cells [126]. However, further functional studies are warranted in order to validate this possibility.

### 4.2. HPV Infection Could Promote EBV Latency Establishment and Lytic Cycle Activation in Head and Neck Epithelial Cells

Cultures of normal epithelial cells in which the EBV lytic cycle is preferentially established appear unable to sustain latent EBV infection [51,127,128]. For instance, Temple et al. (2014) demonstrated that EBV establishes a predominantly productive infection in organotypic cultures of primary oral keratinocytes, characterized by the expression of latent cycle proteins (e.g., EBNA1, LMP1, EBNA2) and lytic cycle proteins (e.g., gB, gp350). Nevertheless, cells expressing exclusively latent proteins were not detected [23]. It is well-known that EBV lytic infection occurs in normal differentiated oral cells from immunosuppressed patients, causing oral hairy leukoplakia (OHL) [129]. Although the presence of EBV latent infection was reported in morphologically normal mucosa, these samples corresponded to tongue and gingival fibrous overgrowths, which are characterized by abnormal enlargement of tissues [51]. Accordingly, it was suggested that normal epithelial cells from the periodontium could be latently infected by EBV [130], but patients considered as “healthy donors” displayed mild periodontal disease (clinical attachment level 1 ≤ x < 3 mm) [131,132]. Additionally, the absence of EBER1 mRNA expression was observed in the normal epithelium adjacent to oral dysplasia [116]. Indeed, an EBV latent infection can be established in nasopharyngeal cells displaying previous genetic alterations, such as hTERT, cyclin D1, or Bmi-1 overexpression, as well as p16 inactivation [133,134,135]. These molecular changes are commonly detected in dysplastic epithelial cells [136,137], which are also more susceptible to EBV infection [116]. Interestingly, the pooled prevalence of HPV infection was 27.2% in oral epithelial dysplasia [138], whereas other authors found a 50% positivity in oral cavity and oropharyngeal dysplasia [139]. Guidry and Scott (2018) reported a significant reduction in EBV DNA replication in both HPV-positive and E7-immortalized human keratinocytes when compared with the EBV DNA replication observed in primary keratinocytes cultured in organotypic rafts [140], suggesting the contribution of HPV to the establishment of EBV latency. Later, a significant decrease in the expression of both EBV immediate early (BZLF1 and BRLF1) and early (BALF5 and BMRF1) genes was identified in HPV-positive tonsillar epithelial rafts when compared with HPV-negative rafts. In addition, a significant increase in the expression of EBER1 (a noncoding RNA highly expressed in latently EBV-infected cells) was evidenced in HPV-positive rafts compared with HPV-negative rafts [141]. On the other hand, the differentiation-dependent cellular transcription factors KLF4 and BLIMP1 can activate BZLF1 and BRLF1 promoters, inducing a differentiation-dependent EBV lytic infection [142]. However, HPV16 E6 and E7 proteins reduce the expression levels of some late differentiation markers, such as FLG, LOR, and WNT5A (which are KLF4 transcriptional targets), blocking EBV replication in human foreskin keratinocytes [141]. Previous studies demonstrated the capacity of TGF-ꞵ/Smad signaling to activate the EBV latent-lytic switch, promoting BZLF1 gene expression [143,144]. Regarding this issue, HPV16 E7 can downregulate TGF-βRII expression in a transgenic mouse model [145]. HPV16 E5 also downregulates the expression of TGF-βRII, which disrupts TGFβ1/Smad signaling in human epithelial cells [146]. Additionally, HPV18 can stabilize the maintenance of EBV genomes in monolayer cultures of oral epithelial cells [147]. Interestingly, HPV integration, a very important event during HPV-driven carcinogenesis, has been related to EBV infection. In fact, EBV presence was associated with a seven-fold (OR 7.11, 95% CI 1.70–29.67) increased probability of the occurrence of HPV16 genome integration [148]. Moreover, EBV infection was found to increase the probability of HPV16 genome integration in the host genome by five-fold (OR 5, 95% CI: 1.15–21.8) [149]. Taken together, these findings suggest the possibility that HR-HPV infection and HR-HPV genome integration can promote EBV latency establishment, which is a very important initial event during EBV-driven carcinogenesis. Meanwhile, an increased percentage of oral cells expressing the Zta protein was evidenced in EBV/HPV18-coinfected raft cultures when compared with HPV18-negative raft cultures [147]. This fact suggests the capacity of HPV18 to promote EBV lytic reactivation in the suprabasal layers of raft cultures, which was also associated with E6 and E7 expression [147].

### 4.3. Immune Evasion Orchestrated by HPV Facilitates the EBV Second Infection

Both HPV and EBV demonstrate a broad range of mechanisms to evade both the innate and adaptive antiviral host response for virus replication (reviewed in [150,151]). Interestingly, some mechanisms of immune evasion in HPV infection are also used by EBV. This fact leads us to hypothesize that primary HPV infection in head and neck epithelial cells could create a more favorable environment for EBV secondary infection. In the following sections, we will describe the mechanisms by which both HPV and EBV evade the immune response to their infection. In general, that response involves the following steps: first, toll-like receptors (TLRs) recognize pathogen-associated molecular patterns (PAMPs) to initiate the antiviral innate immune response [152,153]; next, the nuclear factor kappa light chain enhancer of activated B cells (NF-κB) and interferon regulatory factors (IRFs) are activated, inducing the expression of IFNs and proinflammatory cytokines.

#### 4.3.1. Modulation of Toll-Like Receptors (TLRs)

An increased expression of TLR3, TLR7, and TLR8 has been associated with HPV16 clearance/control in cervical specimens [154]. However, HPV16 can inhibit TLR2, TLR3, TLR7, TLR8, and TLR9 expression, thus contributing to viral persistence in the cervical mucosa [155]. In HPV-positive OPSCCs, TLR5 and TLR9 expression was reduced compared to HPV-negative tumors [156]. In addition, the expression of HPV16 E6 and E7 oncoproteins was found to be related to TLR9 downregulation at both mRNA and protein levels, whereas HPV6 E6 and E7 proteins were unable to repress the activity of the TLR9 promoter [157]. This finding was also associated with the capacity of HPV16 E7 to induce the recruitment of an inhibitory NF-κBp50–p65 complex in the TLR9 promoter, suggesting that TLR9 is involved in cervical carcinogenesis [158]. EBV also disrupts the TLRs sensing in order to modulate the host´s innate immune response. For instance, BGLF5, an alkaline exonuclease expressed during the EBV lytic phase, decreases the expression of TLR2 and TLR9 [159,160]. The large tegument protein BPLF1 also suppresses the TLR2-mediated activation of NF-κB signaling, as well as IL-18 production [161]. Furthermore, EBV can downregulate the expression of TLR9 through LMP1-mediated NF-κB activation [162]. However, LMP1 induces TLR3 expression, leading to an increased NF-κB p65 signaling activation in NPC cells [163]. Interestingly, the expression of nuclear TLR2 and TLR5 was significantly decreased in EBV-positive NPC tumors when compared with HPV-positive and EBV/HPV-negative groups [164]. Since TLR2, TLR3, TLR7, and TLR9 are able to sense EBV infection by monocytes and plasmacytoid dendritic cells, consequently activating the host antiviral response [152,153,165,166], it could be hypothesized that TLR downregulation promoted by HPV infection may disrupt the innate immune recognition of EBV, thus facilitating this infection. In this regard, a polymorphism in the TLR9 1635 locus, related to the HIV acquisition, was also associated with an increased risk of EBV infection [167]. Finally, single nucleotide polymorphisms (SNPs) in TLR4 and TLR9 genes were associated with an increased risk of infectious mononucleosis (IM), an acute disease typically caused by EBV infection [168].

#### 4.3.2. Regulation of IRF Signaling and IFNs Production

IFNs constitute one of the first lines of host defense against virus infections. The production of IFN-α and IFN-β is activated mainly by the IRFs 3 and 7 (reviewed in: [169]). The repression of IRF and Type I IFN production is a mechanism used by HR-HPV to disrupt an effective antiviral response. Specifically, HPV16 E6 binds to IRF3, inhibiting the expression of IFN-β [170]. Additionally, E6 can decrease the expression of IFN-α [171]. A role was also reported for HPV16 E5 in the suppression of stromal Type I IFNs signaling, which involved the function of IRF3 and IRF7 [172]. The capacity of HR-HPVs to inhibit the expression of IFN-stimulated genes (ISG), such as IFIT1, MX1, STAT1, RIG-I, and MDA5, consequently impairing the production of IFN-β and IFN-λ, has previously been reported [173]. Additionally, HPV16 E6 can reduce IFN-κ at mRNA and protein levels in human keratinocytes [173,174,175]. The production of IFN-κ is also decreased by HPV16 and HPV18 E2 through STING downregulation [176]. Similarly, the immediate-early EBV BZLF1 and BRLF1 proteins reduce the activation of IRF3 and IRF7, decreasing the synthesis of IFN-α and IFN-ꞵ [177,178]. BZLF1 also reduces the activation of the IRF1 and STAT1 phosphorylation mediated by IFN-γ [179]. LF2 (BILF4) inhibits IRF7, leading to the suppression of IFN-α production [180], whereas BGLF4 interferes with IRF3 activity, reducing the synthesis of IFN-β [181]. The microRNA (miR)-BART16-5p downregulates the expression of the CREB-binding protein (CBP), a transcriptional coactivator in IFN signaling, abrogating the production of ISGs (IFIT1 and ISG15) in response to IFN-α stimulation [182]. Finally, the miR-BART6-3p suppresses the IFN-β production, targeting the RIG-I-like receptor [183].

#### 4.3.3. Interference with Other Innate Effector Molecules

The host innate response to viral infection is composed of the production and release of pro-inflammatory cytokines, which are also hijacked by viral proteins. For instance, HPV E2, E6, and E7 oncoproteins induce the production of interleukin-10 (IL-10), which, in turn, activates the Janus kinase-1/tyrosine kinase-2/signal transducer and activator of transcription 3 (JAK1/Tyk2/STAT3). The stimulation of the STAT3 pathway inhibits macrophage activation and the production of some inflammatory cytokines, such as TNF-α, IL-6, and IL-1β [184]. Other pro-inflammatory cytokines, such as IL-2 and IL-23, are downregulated in HPV-positive cervical lesions [185]. HPV16 E6 induces the downregulation of the pleiotropic IL-18 [186], which also induces the production of IFN-γ [187]. Reduced levels of IL-1β and IL-18 are associated with an increased risk of developing cervical cancer [188]. HPV E6 reduces the mRNA and protein levels of GM-CSF and TNF-α, interfering with the NF-κB signaling pathway [189]. Similarly, the EBV BCRF1 gene encodes for viral interleukin-10 (vIL-10), a homolog of human IL-10. vIL-10 inhibits IFN-γ, TNF-α, and GM-CSF production by monocytes [190]. BARF1 acts as a soluble receptor for the macrophage colony-stimulating factor (M-CSF), reducing the differentiation and activation of macrophages and inhibiting both the synthesis and release of IFN-α [191,192]. Moreover, BARF1 induces the downregulation of other human cytokines, such as IL-1α and IL-8 [193]. The expression of IL-1α on the cell membrane was found to be related to an anti-tumor immune response [194]. The EBV miR-BHRF1-2-5p targets IL-1 receptor 1 (IL1R1), blocking the activation of NF-κB mediated by IL-1β [195]. HPV16 E6 reduces the level of IL-1β by the proteasomal degradation of pro-IL-1β [196]. However, both tumor-inhibiting and tumor-promoting effects have been reported for IL-1β. This cytokine activates the Th1 CD4+ T cells, which, in turn, secrete IFN-γ and TNF-α inflammatory cytokines [197,198].

#### 4.3.4. Disruption of NF-κB Signaling Pathway

It has been reported that HPV16 E7 decreased the activity of the IKK complex (α and β), which induces the IκBα phosphorylation and degradation of the NF-κB inhibitor [199]. Furthermore, HPV16 E6 interferes with the NF-κB RelA/p65-dependent transcriptional activity, cooperating with the disruption of TNFα-induced NF-κB signaling [199]. HPV16 E6 and E7 increase the p105 (NFKB1) and p100 (NFKB2) levels, thus inhibiting the transcriptional activity of NF-κB complexes [200]. E6 also inhibits the activation of NF-κB induced by the CBP/p300 coactivator [201], although both HPV E6 and E7 bind to CBP/p300 (a p65 transcriptional coactivator) and p300/CBP-associated factor (P/CAF), disrupting IL-18 transcription [202]. Additionally, EBV genes are involved in disrupting NF-κB signaling. For instance, BZLF1 suppresses the NF-κB pathway, inducing p65 nuclear translocation but disrupting its transcriptional function [203]. Additionally, BPLF1 deubiquitinates the TNF receptor-associated factor 6 (TRAF6), inhibiting NF-κB signaling during lytic infection [161]. EBNA1 inhibits the phosphorylation of IKKα/β, disrupting the canonical NF-κB pathway [204]. EBV BGLF4 downregulates NF-κB transactivation by disrupting the interaction of the NF-κB coactivator UXT and p65 [205]. BGLF2 downregulates TNFα-induced NF-κB activity, blocking the nuclear translocation of p65 [206]. Taken together, these facts suggest that both HPV and EBV have the capacity to disrupt the antiviral immune response in the head and neck region, which potentially facilitates secondary EBV infection. Interestingly, both HPV and EBV-encoded proteins are also able to activate the NF-κB signaling pathway, promoting cancer progression [207,208]. Indeed, NF-κB is a pivotal link between chronic inflammation and cancer [209]. Conversely, it was suggested that NF-κB has tumor suppressor properties after HPV infection, and that its downregulation by E6/E7 viral proteins contributes to cervical cell immortalization [210].

#### 4.3.5. Downregulation of Major Histocompatibility Complex (MHC) by HPV Oncoproteins

It has been reported that the HR-HPV E5 oncoprotein can disrupt antigen presentation by downregulating MHC/HLA class I membrane expression with accumulation in the Golgi’s apparatus [211,212]. This effect results in a reduced recognition of HPV-infected cells by CD8+ T cells [212]. Additionally, the HR-HPV E7 oncoprotein has been shown to repress the MHC class I heavy chain promoter [213], contributing to escaping from immune surveillance.

### 4.4. Other Mechanisms Involved in EBV/HPV-Associated HNCs

It has been proposed that HR-HPV and EBV interactions could contribute to cancer EMT and cancer progression, including HNC [214,215]. However, mechanistic studies that are focused on the role of EBV/HPV co-infection and HNC progression are scarce. In this section, we describe additional pathways and mechanisms involved in a potential cooperation between both viruses for HNC development.

#### 4.4.1. Cooperation between HR-HPV E6 and EBV LMP1

Al-Thawadi et al. found an association between the LMP1/E6 co-expression and upregulation of the inhibitors of DNA binding 1 (Id-1) in cervical carcinomas [216]. Indeed, LMP1 has been reported to induce Id-1 by inactivating the Foxo3a function in nasopharyngeal cells [217]. Id-1 can activate the NF-κB/survivin and phosphoinositide 3-Kinase/Akt signaling pathways, promoting HNC cell growth [218]. Additionally, LMP1 alone, or in combination with HPV16 E6, was able to induce the expression of γH2AX, which is considered a DNA damage biomarker. Moreover, a HPV16 E6 and LMP1 copresence reduces the expression of p53, RB, p27, and ChK1, suggesting a synergism between E6/LMP1 proteins and DDR [219]. Similarly, under genotoxic conditions, a reduction in DDR proteins, such as ATM, ATR, Chk1, and Chk2, as well as an increased resistance to apoptosis, was evidenced in HPV16 E6+/EBV LMP1+ cells compared to those expressing only the HPV16 E6 protein [220]. In fact, the expression of both HPV16 E6 and EBV LMP1 was able to increase the proliferation rates of MEFs cells compared to those cells expressing the HPV16 E6 oncoprotein alone, which was associated with NF-κB activation and p53 suppression [219,220]. In addition, increased soft agar colonies, as well as tumor formation in nude mice, were evidenced when HPV16 E6+/EBV LMP1+ cells were compared to mock cells. This finding was associated with increased levels of AKT and adhesion molecules (MMP2, CAT-1, and paxillin) in HPV16 E6 +/ EBV LMP1+ cells [220]. However, no effect in the proliferation rates was evidenced when MEFs expressing EBNA1 or EBNA1/E6 were analyzed [219]. The expression of HR-HPV E6 and EBV LMP1 oncoproteins was associated with grade 3 infiltrating ductal carcinoma, as well as with the occurrence of one or more lymph node metastasis when compared with HPV-, EBV-, or HPV-/EBV- tumors [221].

#### 4.4.2. Gene Promoter Methylation Promoted by HR-HPV/EBV Infection

An increased frequency of TP53 promoter methylation was evidenced in HR-HPV/EBV-infected OPSCCs compared to HR-HPV or EBV infection alone [222]. Additionally, HPV/EBV coinfection was associated with the E-cadherin gene (CDH1) and RB1 promoter methylation in cervical lesions [223]. However, HR-HPV/EBV coinfection was not associated with the methylation status of the death-associated protein kinase (DAPK) gene in HSILs [224]. DAPK is a family of serine/threonine kinases with tumor suppressor functions that operate by regulating apoptosis and autophagy, which are frequently downregulated in cancer by gene promoter methylation [225].

#### 4.4.3. Reduced Activity of Detoxifying Enzymes Promoted by HR-HPV/EBV Infection

In OPSCC, the activity of glutathione peroxidase (GPx) and superoxide dismutase (SOD) antioxidant enzymes was significantly decreased in patients with HPV/EBV coinfection when compared with HPV or EBV infection alone. A low GPx and SOD activity was observed in HPV/EBV-coinfected patients with poorly differentiated tumors and an advanced stage of disease [226].

#### 4.4.4. Gene Polymorphisms and HR-HPV/EBV Coinfection

Sharma et al. (2019) studied the association of the TLR 9 (−1486 T/C) genotype, and wild (TT), hetero (CT), and homozygous (CC) allele, with HPV/EBV coinfection in pre-malignant and OSCC samples [227]. Interestingly, in HPV+/EBV+ patients, the occurrence of the TLR 9 (−1486 T/C) hetero and homozygous (CT + CC) genotype was associated with a 22 (OR 22.5, 95% CI: 1.608 to 314.77) and 30 (OR 30.0, 95% CI: 2.188–411.25) times increased probability of pre-malignant and OSCC development, respectively [227]. An association was also identified between TLR 4 (+896A/G) polymorphism and the development of oral premalignant lesions in HPV/EBV-coinfected patients [227]. TLR polymorphisms induce modification in both cytokines and chemokines patterns, thus increasing the probability of NPC development [228,229]. Furthermore, evidence suggested that high levels of EBER1 may contribute to HPV-mediated oncogenesis by modulating both IFN-related genes and pro-inflammatory cytokine genes [230]. A hypothetical model of HR-HPV/EBV cooperation is shown in Figure 1.

## 5. Conclusions and Remarks

Single HR-HPV or EBV infections are insufficient for carcinogenesis, suggesting the involvement of additional factors. Both HR-HPV and EBV have been detected in HNCs, most notably in oropharyngeal tumors. The HPV/EBV copresence in HNCs is highly variable worldwide, ranging from 6.5% to 37.5% of OSCCs, 15% to 47.7% of NPCs in endemic regions, and 53.3% to 86% of OPSCCs. Potential mechanisms for HPV/EBV cooperation remain unclear, although some possibilities have been suggested. First, HPV could facilitate EBV entry by promoting an increased expression of integrins and CD21. Second, HPV could facilitate EBV latency establishment by promoting genetic alterations due to HR-HPV genome integration into the host genome. Third, local immune evasion promoted by HPV infection could facilitate secondary EBV infection in epithelial cells. Additionally, HPV and EBV viral oncoproteins can synergistically cooperate for head and neck carcinogenesis. Additional studies are warranted in order to evaluate the mechanisms involved in the cooperation between HPV and EBV for head and neck carcinogenesis.

## Figures and Tables

**Figure 1 biology-10-01232-f001:**
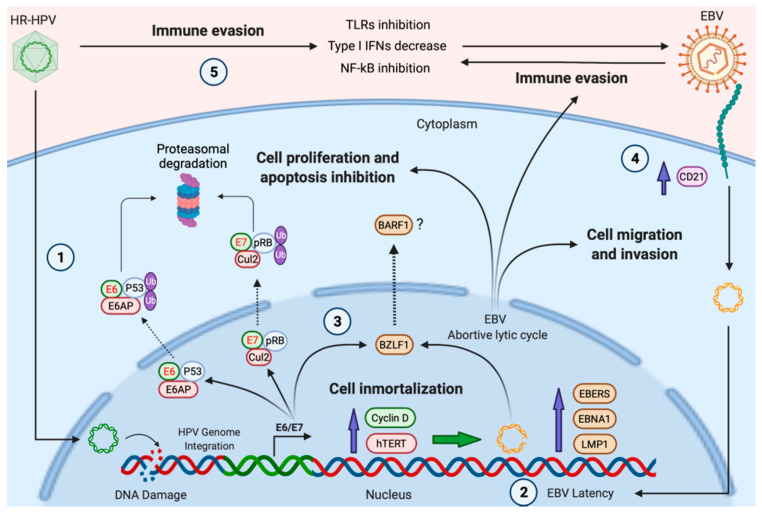
A hypothetical model of HPV and EBV cooperation for the development of HNCs. (**1**), HR-HPV infection and viral genome integration promote DNA damage in head and neck epithelial cells; (**2**), previous DNA alterations, as well as the increased expression of cyclin D1 and hTERT induced by HR-HPV, can promote the establishment of EBV latency; (**3**), HR-HPV E6/E7 oncoproteins induce the expression of BZLF1, favoring the expression of EBV lytic genes with oncogenic properties, such as BARF1 (abortive lytic cycle); (**4**), HR-HPV infection induces CD21 (CR2), which, in turn, facilitates EBV cell entry; (**5**), the expression of HR-HPV oncoproteins inhibits protagonists of the anti-viral immune response, inducing immune evasion. Green shapes with red text represent HR-HPV oncoproteins. Brown shapes symbolize EBV proteins. Created by BioRender.com (accessed on 5 November 2021).

**Table 1 biology-10-01232-t001:** Comparison between HPV and EBV epithelial cell entry.

Biological Process	Human Papillomavirus	Epstein–Barr Virus	References
Route of entry	Direct epithelial contact, apical entry (microlesions)	Salivary transmission, apical, basolateral, or basal entry	[21,33]
Tropism	Epithelial cells (mucosal or cutaneous)	Epithelial cells (mucosal), B cells, T cells, NK cells	[33,34]
Entry mechanism	Endocytosis	Membrane fusion	[35,36]
Receptors	Entry receptor complex, HSPGs, integrins	EphA2	[24,37]

HSPGs, heparan sulphate proteoglycans; EphA2, ephrin receptor tyrosine kinase A2.

**Table 2 biology-10-01232-t002:** HPV prevalence in HNSCC according to meta-analysis data.

Tumor Type	No. of Cases	HPV Pooled Prevalence (%)	OR (95% CI)	Comments	Refs
OSCC	4680	46.5	37.6–55.5	HPV infection was 4.7 times more likely to be detected in OSCC than in normal mucosa.	[106]
OSCC	610	58.0	54.1–61.9	HPV positivity increased the probability of OSCC development by 12.7 times.	[44]
OSCC HNSCC (other sites)	3238 4852	38.1 34.5	30.0–46.2 28.4–40.6	The pooled HPV prevalence was greater in OSCC (38.1%) than in other HNSCCs (24.1%).	[43]
NPC	1748	21.0	1.69–13.45	HPV prevalence was higher in cases outside of China than in cases from regions in China (23% vs. 19%; *p* < 0.001).	[85]
NPC	2453	19.9	13.6–27.1	HR-HPV infection was higher in WHO type I NPCs (39.9%) compared to WHO types II/III tumors (23.3%).	[77]
OPSCC	6009	44.8	36.4–53.5	HPV pooled prevalence was more increased in New Zealand (74.5%), Sweden (70.0%), and Denmark (61.7%) than in Brazil (11.1%), Germany (25.0%), and the Netherlands (30.3%).	[98]
OPSCC HNSCC (all sites)	925 5681	41.0 21.9	38.0–44.0 21.0–23.0	HR-HPV (any genotype) and HPV16 increased the probability of HNSCC development by 1.83 and 4.44 times, respectively.	[107]
OPSCC HNSCC (other sites)	5396 13,972	47.7 21.8	42.9–52.5 18.9–25.1	-	[108]
OSCC OPSCC LSCC HNSCC (all sites)	2642 969 1435 5046	23.5 35.6 24.0 25.9	21.9–25.1 32.6–38.7 21.8–26.3 24.7–27.2	HPV prevalence was significantly higher in OPSCC than OSCC or LSCC.	[109]
OSCC OPSCC LSCC/HPSCC HNSCC (all sites)	5478 3946 2739 12,163	24.2 45.8 22.1 29.5	18.7–30.2 38.9–52.9 16.4–28.3 25.5–33.6	HPV16 prevalence was more increased in OPSCC (40.6%) than in OSCC (14.9%) or LSCC (13.4%).	[102]
OSCC OPSCC LSCC HNSCC (all sites)	3153 2768 856 6777	37.5 40.5 23.6 37.0	35.9–39.2 38.7–42.3 22.1–25.0 36.0–38.0	The association of HPV with cancer development was increased for OPSCC (OR: 14.7) compared to OSCC (OR: 4.1) or LSCC (OR: 3.2).	[110]
LSCC	2559	28.0	23.5–32.9	HPV infection was significantly associated with the risk of LSCC development (OR: 5.4).	[111]

HNSCC, head and neck squamous cell carcinoma; OSCC, oral squamous cell carcinoma; OPSCC, oropharyngeal squamous cell carcinoma; LSCC, laryngeal squamous cell carcinoma; HPSCC, hypopharyngeal squamous cell carcinoma; NPC, nasopharyngeal carcinoma.

**Table 3 biology-10-01232-t003:** EBV prevalence in HNSCCs.

Tumor Type	Method	No. of Cases	EBV Positivity (%)	Comments	Refs
OSCC	EBV-chip hybridization	57	82.5	-	[48]
OSCC	PCR for BNLF1	91	45.1	EBV was significantly associated with the probability of OSCC development (OR 3.76).	[49]
OSCC	EBER in situ hybridization	165	41.2	-	[50]
NPC	EBER in situ hybridization	92	57.6	-	[72]
NPC	EBER in situ hybridization	62	85.5	-	[73]
NPC	EBER in situ hybridization	150	62.0	Overall survival was increased in EBV-positive patients (*p* = 0.005).	[74]
NPC	EBER in situ hybridization	19	84.2	-	[75]
NPC	EBER in situ hybridization	82	87.8	EBV positivity was evidenced in 92.6% of non-keratinizing carcinoma.	[76]
NPC	EBER in situ hybridization	56	73.2	EBV infection was not associated with the histopathological type of NPC.	[112]
NPC	-	2329	76.7	Meta-analysis. EBV prevalence was increased in WHO Type II/III (83.2%) compared to WHO Type I cases (21.3%).	[77]
OPSCC	Nested PCR	62	29.0	-	[56]
OPSCC	PCR-ELISA	28	85.7	-	[103]
OSCC NPC OPSCC HPSCC LSCC	EBER in situ hybridization	37 20 74 50 28	0 60.0 1.4 0 0	EBER positivity was significantly higher in NPCs compared to non-NPC HNSCCs (*p* < 0.001).	[113]
HNSCC (OSCC, PSCC, and LSCC)	PCR for LMP1	98	69.4	LMP1 protein was also expressed in PSCC (100%) followed by OSCC (76.0%) and LSCC (33.3%).	[10]

HNSCC, head and neck squamous cell carcinoma; OSCC, oral squamous cell carcinoma; OPSCC, oropharyngeal squamous cell carcinoma; LSCC, laryngeal squamous cell carcinoma; HPSCC, hypopharyngeal squamous cell carcinoma; NPC, nasopharyngeal carcinoma; PSCC, pharyngeal carcinoma.

## Data Availability

Not applicable.

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
