# Peer review of "High-Risk Human Papillomavirus and Epstein–Barr Virus Coinfection: A Potential Role in Head and Neck Carcinogenesis"

_biology, 2021, doi:10.3390/biology10121232_

Round 1
Reviewer 1 Report
The authors addressed all my comments
Author Response
Thank you very much. The english grammar was extensively checked.
Reviewer 2 Report
I understand that there is insufficient direct evidence for HPV and EBV co-infection. A review of this subject must be rooted on hypothetical interactions.
Thank you for providing an update.
Author Response

(The authors gave the same response as above.)

Reviewer 3 Report
l am satisfied with the responses to my comments on the first submission, and have only one additional request: HPV 16 E5's role in the suppression of MHC class I expression should be added to the now more exhaustive section on immune suppresion.
Author Response
Thank you very much. HPV 16 E5's role in the suppression of MHC class I expression was added.
This manuscript is a resubmission of an earlier submission. The following is a list of the peer review reports and author responses from that submission.
Round 1
Reviewer 1 Report
In this paper Blanco et al. review the co-infection of high-risk HPVs and EBV and their potential mechanisms in human head and neck cancers (HNCs). They highlighted three potential mechanisms of HPV and EBV crosstalk in HNCs which are: First, HPV can facilitate EBV entry by promoting increased expression of integrins and CD21. Second, HPV could help EBV latency establishment by promoting genetic alterations due to HPV genome integration into the host genome cell. Third, local immune evasion promoted by HPV infection can accelerate the secondary EBV infection in epithelial infected cells. Well, this is a very interesting review paper about HPV and EBV oncoviruses coinfection and their mechanism in human HNCs. Thus, I believe that the paper can be accepted for publication after minor revisions as follows:
- “High-risk HPVs” should be mentioned in the title since the review focuses on high-risk HPVs
- The authors should add a paragraph about HPV and EBV oncoproteins signaling pathways, especially beta-catenin, and their involvement in the induction and amplification of the EMT event which is an important hallmark of cancer progression.
- Authors should add 1-2 tables with refs in which they can summarize the meta-analysis studies of HPV and EBV in HNCs
- The paper needs some revisions for English
Author Response
Reviewer 1. In this paper Blanco et al. review the co-infection of high-risk HPVs and EBV and their potential mechanisms in human head and neck cancers (HNCs). They highlighted three potential mechanisms of HPV and EBV crosstalk in HNCs which are: First, HPV can facilitate EBV entry by promoting increased expression of integrins and CD21. Second, HPV could help EBV latency establishment by promoting genetic alterations due to HPV genome integration into the host genome cell. Third, local immune evasion promoted by HPV infection can accelerate the secondary EBV infection in epithelial infected cells. Well, this is a very interesting review paper about HPV and EBV oncoviruses coinfection and their mechanism in human HNCs. Thus, I believe that the paper can be accepted for publication after minor revisions as follows:
“High-risk HPVs” should be mentioned in the title since the review focuses on high-risk HPVs
Answer: Many thanks for this observation. This was corrected.
Reviewer 1. The authors should add a paragraph about HPV and EBV oncoproteins signaling pathways, especially beta-catenin, and their involvement in the induction and amplification of the EMT event which is an important hallmark of cancer progression.
Answer: Many thanks for this comment. Indeed, Gupta I et al. (2021), published a manuscript in which a cooperation between HPV and EBV for enhancing cell motility and invasiveness by beta catenin pathway in breast cancer cells. However, this manuscript was recently retracted (https://pubmed.ncbi.nlm.nih.gov/34557421/). Thus, we deleted this information from this review. However, extensive information is available about HPV or EBV role activating beta catenin pathway, and a very nice review about this topic was published in 2019 by Cyprian FS et al.
Reviewer 1. Authors should add 1-2 tables with refs in which they can summarize the meta-analysis studies of HPV and EBV in HNCs.
Answer: Tables summarizing the HPV and EBV meta-analysis were included (Tables 2 and 3).
Reviewer 1. The paper needs some revisions for English
Answer: The manuscript was extensively checked by a native English speaker.
Reviewer 2 Report
The submitted manuscript reviewed the biological changes after HPV or EBV infection and proposed possible interactions under HPV/EBV coinfection.
Major:
This manuscript would be a great grant application instead of a review article. In my opinion, a review article should summarize what we have known about HPV and EBV coinfection rather than a series of hypotheses of HPV and EBV interactions. Publishing bioinformatics analysis results in a review article is also not proper since there is no methods section in a review article.
Author Response
Reviewer 2.
The submitted manuscript reviewed the biological changes after HPV or EBV infection and proposed possible interactions under HPV/EBV coinfection.
Major:
This manuscript would be a great grant application instead of a review article. In my opinion, a review article should summarize what we have known about HPV and EBV coinfection rather than a series of hypotheses of HPV and EBV interactions. Publishing bioinformatics analysis results in a review article is also not proper since there is no methods section in a review article.
Answer: Many thanks for these observations and comments. This manuscript summarizes published epidemiological information about HR-HPV/EBV copresence in HNCs. This is not a closed problem, in fact, is unclear a cooperation between HR-HPV/EBV in a subset of HNCs. However, considering published evidence, we suggest potential mechanisms of HR-HPV/EBV cooperation, although these possibilities require additional experimental confirmation. This was clarified into the manuscript. Additionally, sentences which are not related to the role of EBV/HPV copresence in HNC were deleted. Bioinformatic analyses come from public data from the Cancer genome Atlas consortium. Thus, the authors only evaluated potential associations using these public data and the methodological information was included in the legend of the figure.
Reviewer 3 Report
This is an excellent review of HPV and EBV coinfection and its possible roles in head-and-neck carcinogenesis. It includes detailed, accurate summaries of many important papers in the field. The inclusion of novel analyses of TCGA data and functional association networks is commendable.
Specific comments:
The T-tests in Tables 2 and 3 should be corrected for multiple comparisons.
Line 22: Recommend deleting "Therefore" which isn't logical here.
Lines 58-60: Reference 13 does not address permissiveness in the upper highly differentiated layers.
Line 76: Recommend deleting "On the other hand." What follows does not contrast with the previous sentence.
Lines 78-79: The data in references 23 and 24 do not provide evidence that EphA2 increases the efficiency of EBV infection; they merely show a correlation. Please reword this sentence to avoid overstating the conclusions.
Line 117: "found 38.1% of HPV DNA" should be corrected to, for example "found 38.1% of cancers contained HPV DNA"
Line 188: "4.852" should be "4852"
Line 193: "which higher" should be "with higher"
Lines 226-229: This description of the results presented in references 104-107 is too brief for those who don't already know the results; please clarify. A simple solution might be to move the sentence in lines 226-7 after the sentence in lines 230-231.
Lines 268-269: Rearrangement of this sentence would help to clarify its meaning, for ex: "Cultures of normal epithelial cells in which the EBV lytic cycle is preferentially established appear unable to sustain latent EBV infection."
Section 4.3: The introductory part of this paragraph isn't logical. If HPV and EBV have ways of evading the immune system, why do they need HIV to suppress the immune system for them? One way to frame this might be: It is critical for viruses to suppress the immune system, and they have evolved to do so, but their methods of evading the immune system are imperfect – it helps them to take advantage of other viruses' prior/ongoing suppression of the immune system.
Rearrangement of the second part would also be helpful. For example, follow line 325 with line 327 to end a first paragraph. Then start a new paragraph with line 326. Then introduce lines 328-333 with something like: "In the following sections, we will describe the mechanisms HPV and EBV use to combat the immune response to their infection. In general, that response involves the following steps:"
Line 458: It's not obvious that it's beneficial to both viruses to up or downregulate the same factors. They would seem to benefit more from regulating different, complementary sets of factors. Of course, partial regulation by each virus could be additive. This issue should receive serious discussion – perhaps its own section.
Line 476: What does "alone" mean in this context?
Line 496: "induces" should be "induce"
Line 502: The relevance of DAPK should be explained.
Section 5: This section should be subdivided into multiple paragraphs with unifying themes, as much as possible. One them that seems to apply to lines 513 to 527, for example, is 'effects on grade/progression.'
Author Response
Reviewer 3.
This is an excellent review of HPV and EBV coinfection and its possible roles in head-and-neck carcinogenesis. It includes detailed, accurate summaries of many important papers in the field. The inclusion of novel analyses of TCGA data and functional association networks is commendable.
Specific comments:
The T-tests in Tables 2 and 3 should be corrected for multiple comparisons.
Answer: Many thanks for these observations. Respect the t-Test, this was used to compare two data series in Tables 2 and 3 (now are Tables 4 and 5). This statistical analysis is applied by default for comparisons between two data series (i.e., TLR2 expression in HPV positive and HPV negative cases) when the Xenabrowser platform (TCGA data) is used.
Reviewer 3. Line 22: Recommend deleting "Therefore" which isn't logical here.
Answer: This was done.
Reviewer 3: Lines 58-60: Reference 13 does not address permissiveness in the upper highly differentiated layers.
Answer: Many thanks. The reference was deleted, and two new references were added.
Reviewer 3: Line 76: Recommend deleting "On the other hand." What follows does not contrast with the previous sentence.
Answer: Many thanks; this was done.
Reviewer 3: Lines 78-79: The data in references 23 and 24 do not provide evidence that EphA2 increases the efficiency of EBV infection; they merely show a correlation. Please reword this sentence to avoid overstating the conclusions.
Answer: The reference 22 (now is ref. 24) showed that EphA2 is the EBV receptor in epithelial cells. In the study by Chen J et al (2018, Nature Microbiol) using functional assays in cell lines, it was demonstrated that EphhA2 works as a EBV receptor. A new reference was added, because this very important study was published at the same time in the same journal by two different laboratories. Additionally, the refs. 23 (now is ref. 26) showed an association between EphA2 and progression/prognosis, so this sentence was modified in the manuscript. The sentence citing the ref. 24 was deleted because was wrongly included.
Reviewer 3: Line 117: "found 38.1% of HPV DNA" should be corrected to, for example "found 38.1% of cancers contained HPV DNA"
Answer: This was corrected.
Reviewer 3: Line 188: "4.852" should be "4852"
Answer: This number was changed.
Reviewer 3: Line 193: "which higher" should be "with higher"
Answer: This was changed
Reviewer 3: Lines 226-229: This description of the results presented in references 104-107 is too brief for those who don't already know the results; please clarify. A simple solution might be to move the sentence in lines 226-7 after the sentence in lines 230-231.
Answer: Many thanks for this observation. We moved the sentence in lines 226-7 after the sentence in lines 230-231.
Reviewer 3: Lines 268-269: Rearrangement of this sentence would help to clarify its meaning, for ex: "Cultures of normal epithelial cells in which the EBV lytic cycle is preferentially established appear unable to sustain latent EBV infection."
Answer: Many thanks for this comment. The sentence was changed according to this observation (lines 293 -294).
Reviewer 3: Section 4.3: The introductory part of this paragraph isn't logical. If HPV and EBV have ways of evading the immune system, why do they need HIV to suppress the immune system for them? One way to frame this might be: It is critical for viruses to suppress the immune system, and they have evolved to do so, but their methods of evading the immune system are imperfect – it helps them to take advantage of other viruses' prior/ongoing suppression of the immune system.
Answer: Many thanks for this very important observation. The sentences related to HIV infection were deleted, and we only considered the sentences related to immune evasion orchestrated by HPV and EBV.
Reviewer 3: Rearrangement of the second part would also be helpful. For example, follow line 325 with line 327 to end a first paragraph. Then start a new paragraph with line 326. Then introduce lines 328-333 with something like: "In the following sections, we will describe the mechanisms HPV and EBV use to combat the immune response to their infection. In general, that response involves the following steps:"
Answer: Many thanks. This sentence suggested by the reviewer was adapted and included.
Reviewer 3: Line 458: It's not obvious that it's beneficial to both viruses to up or downregulate the same factors. They would seem to benefit more from regulating different, complementary sets of factors. Of course, partial regulation by each virus could be additive. This issue should receive serious discussion – perhaps its own section.
Answer: The section was completely reorganized, many thanks for this observation.
Reviewer 3: Line 476: What does "alone" mean in this context?
Answer: The sentences in lines 471-477 were deleted because are related to other types of cancer.
Reviewer 3: Line 496: "induces" should be "induce"
Answer: This was changed.
Reviewer 3: Line 502: The relevance of DAPK should be explained.
Answer: The following sentence was added to the manuscript (lines 521-523): “DAPK is a family of serine/threonine kinases with tumor suppressor functions through regulating apoptosis and autophagy, which are frequently downregulated in cancer by gene promoter methylation”
Reviewer 3: Section 5: This section should be subdivided into multiple paragraphs with unifying themes, as much as possible. One them that seems to apply to lines 513 to 527, for example, is 'effects on grade/progression.'
Answer: This section was completely reorganized, considering new subsections; the themes were unified.
Round 2
Reviewer 2 Report
Thanks for the update.